# Hierarchical Prompts with Context-aware Calibration for Open-Vocabulary Object Detection

## Abstract

Open Vocabulary Object Detection (OVD) aims to extend to identify novel classes solely through text descriptions, by learning the mapping between images and text from the base class. However, current methods focus on linking the visual features of target objects to their corresponding category names for prompt learning, ignoring richer contextual information and shared knowledge about these categories, which can easily lead to overfitting on base categories and exhibit poor generalization to novel classes. To address the above problems, we propose **Hi**erarchical prompts with **C**ontext-**A**ware calibration (HiCA) for open-vocabulary object detection, which integrates high-level semantic and contextual information into the detector from both linguistic and visual perspectives. Hierarchical prompts effectively map regions with superior-level semantics, which encompasses shared knowledge of both base and novel classes, thereby enhancing the model's generalization ability to novel classes. Meanwhile, context-aware calibration utilizes the visual context of the image to establish the correlation between contextual information and categories, thereby minimizing the adverse effects of the background and enhancing generalization to novel classes. Extensive experiments demonstrate that the hierarchical prompts with context-aware calibration can effectively improve the performance of the open vocabulary detection methods. Especially on the OV-COCO, we achieve $57.2\%$ $\text{mAP}_B$, surpassing the current state-of-the-art by $2.4\%$ while achieving the best $\text{mAP}_{50}$.

## 1 Introduction

In recent years, advancements in deep neural networks have propelled significant progress in object detection (He et al., 2017; Lin et al., 2017; Ren et al., 2015), leading to its broad application across various downstream tasks (Li et al., 2023; Ma et al., 2023). However, most of the existing studies have focused on closed-set scenarios that necessitate extensive labeled data. In contrast, a real open-world environment continuously presents novel categories, making the collection and labeling of samples for each novel class increasingly challenging. The challenge significantly limits the practical usability of these methods. To address the issue, Open-Vocabulary Object Detection (OVD) tasks (Zareian et al., 2021; Gu et al., 2022) have been proposed, aiming to recognize novel classes during the testing phase by leveraging semantic embeddings of category names as classifiers. This approach enables the classification of regions into appropriate categories, facilitating the detection of previously unseen objects without the need for extensive retraining. Consequently, OVD enhances the adaptability and generalization of object detection models in dynamic and open-world settings.

By leveraging large-scale pre-trained Vision and Language Models (VLMs), such as CLIP (Radford et al., 2021) and ALIGN (Jia et al., 2021), recent advances in OVD have employed knowledge distillation to transfer the insightful knowledge of VLMs to the object detection task. This approach enables the models to generalize and identify unknown object categories effectively. Substituting traditional classifiers with embeddings of class names and prompts allows for seamless adaptation to emerging classes. The prompts can either be handcrafted, as in RegionCLIP (Zhong et al., 2022) or learnable, as in PromptDet (Feng et al., 2022). Overall, these techniques distill visual features from VLMs into detection frameworks and utilize learnable multi-modal prompts to facilitate knowledge extraction from large models.

Figure 1: Existing methods primarily focus on learning the mapping between visual features of objects and their corresponding class prompts, neglecting to capture shared knowledge between both base and novel classes and rich information in the visual context. This oversight leads to misclassifications, particularly in cases where objects from different superclasses share similar appearances. Consequently, detectors tend to favor the heavily trained base classes, resulting in overfitting and diminished generalization capabilities for novel classes. Our approach introduces high-level semantic information from both linguistic and visual perspectives, to improve the generalization ability of the model.

Current methods primarily focus on learning the mapping between visual features of objects and their corresponding class prompts, neglecting to capture the shared knowledge between both base and novel classes, as well as the rich information present in the visual context. This oversight leads to misclassifications, particularly in cases where objects from different superclasses share similar appearances. Consequently, detectors tend to favor the heavily trained base classes, resulting in overfitting and diminished generalization capabilities for novel classes. Our approach, in contrast, incorporates high-level semantic information from both linguistic and visual perspectives. This integration allows us to leverage coarse-grained and contextual knowledge, ultimately enhancing the overall performance of open-vocabulary detection and ensuring robust generalization of the model across various classes.

Despite the rapid advancements in OVD, several challenges remain. Firstly, as shown in Figure 1, current approaches predominantly map regions to classes to learn the prompts, which only learn the relationship between object regions and base classes during the training process. As a result, the learned prompts tend to favor detecting regions as base classes, exhibiting poor generalization to novel classes. Simultaneously, prevailing methodologies often treat visual context as purely negative examples, inadvertently creating barriers to distinguishing novel categories from background elements. The rich information contained in the context is largely underutilized, as these methods fail to leverage the explanatory power of the context by clearly establishing connections between contextual features and class identifiers. This oversight limits the potential for detailed understanding and adaptive learning in the detection of novel classes.

To address the aforementioned issues, we propose hierarchical prompts with context-aware calibration (HiCA) for open-vocabulary object detection. Our framework enhances semantic and visual alignment with more generalization ability through the integration of high-level languages and visual context. Specifically, hierarchical prompts decompose the original region-category mapping into a two-step process: coarse-grained mapping between objects and superclasses and fine-grained between class and superclasses. The coarse-grained superclass knowledge effectively encompasses both base and novel classes, thereby ensuring that the prompts do not overly favor base classes during the training phase. Furthermore, context-aware calibration fosters a strong connection between contextual information and categories through the context-aware matrix, which obtains visual embeddings of contexts by unsupervised clustering and constructs the context-superclass distribution using a Distribution Generation Layer (DG Layer). Moreover, the association between superclasses and categories is used to fit the context-class distribution. By selecting the appropriate class distribution based on the specific context, we calibrate the detection results, ensuring a more precise and context-sensitive classification.

The effectiveness of HiCA is evaluated on popular open-vocabulary object detection benchmarks OV-COCO and OV-LVIS. We establish a baseline based on a knowledge distillation framework on

OV-COCO and achieve the state-of-the-art performance of 57.2% mAP in the base class and 50.4% mAP in the overall class while ensuring the generalization of novel classes.

## 2 RELATED WORK

**Open-Vocabulary Object Detection** Open-vocabulary object detection has recently become a focus in the modern object detection area, which aims to detect objects of unlimited categories. OVR-CNN (Zareian et al., 2021) is the first work that put forth the OVD task, which pre-trains the detector with image-caption pairs to enable generalization to novel classes. Since pre-trained vision-language models like CLIP (Radford et al., 2021) emerged and demonstrated strong capabilities, various research has incorporated VLMs in their methods. CORA (Wu et al., 2023b) proposed region prompting and anchor pre-matching to tackle the whole-to-region distribution gap and make the object queries class-aware, avoiding the low efficiency of OV-DETR. ViLD (Gu et al., 2022) employs a knowledge distillation approach that aligns the region embeddings of detected objects to visual and text representations inferred by the teacher VLM. SAMP (Zhao et al., 2024) designed a mechanism to generate scene-adaptive and region-aware multi-modal prompts to enhance knowledge transfer. In this paper, we pay more attention to high-level semantic knowledge and introduce coarse-grained information to improve the detection and generalization performance of the model simultaneously.

**Prompt tuning** Prompt tuning has emerged as a significant advancement in natural language processing and found its way into the realm of computer vision, where it has been adapted to enhance performance in tasks such as image classification and object detection. CLIP demonstrated how textual prompts could be paired with images to create a joint vision-language embedding space. This approach allows the model to classify images based on textual descriptions, thereby leveraging large-scale pre-trained language models for visual tasks. CoOp (Zhou et al., 2022a) learns continuous prompt vectors, enabling the model to adaptively modify the prompts for better alignment with the image data. VPT (Jia et al., 2022) prepends a set of learnable parameters to transformer encoders and remarkably beats full fine-tuning on 20 downstream recognition tasks. We use hierarchical prompts to fuse coarse-grained and fine-grained knowledge and construct a multi-modal prompts architecture to further optimize the detector performance.

## 3 METHODS

### 3.1 PROBLEM DEFINITION

In the open-vocabulary object detection, we have a training set $\mathcal{D}^T = \{(I_i, O_i)\}_{i=1}^{|\mathcal{D}^T|}$, where $I_i$ represents input images and $O_i = \{(r_j, y_j)\}_{j=1}^{|O_i|}$ denotes the annotation information. The $r_j \in \mathbb{R}^4$ is the bounding box of the object, and $y_j \in C^B$ is the class to which the object belongs. Here, $C^B$ refers to the base classes. During the testing phase, We define the categories that only appear in the test set $\mathcal{D}^V$ as novel classes $C^N$, with the condition that $C^N \cap C^B = \emptyset$. We define $C^S$ as the set of all superclasses to which the combined set of base and novel classes $C = C^B \cup C^N$ belongs.

### 3.2 OVERALL FRAMEWORK OF HICA

Our knowledge distillation-based open-vocabulary object detection model is built upon the Faster R-CNN (Ren et al., 2015) architecture, which serves as the student model, while the Vision-Language Models CLIP (Radford et al., 2021) acts as the teacher model. Existing methods usually use prompts to reformulate the textual and visual inputs for the teacher model, adapting them to downstream tasks. Specifically, the text encoder $E_t$ of the teacher model computes the category embeddings $\{e_c^t\}_{c \in C} \in \mathbb{R}^d$ using the prompt templates such as "a photo of [class]". Compared to fixed template prompts, learnable prompts significantly minimize the reliance on manual design. These learnable text prompts usually take the form $\{[v_1], [v_2], ..., [v_M], [\text{CLASS}]\}$, where $\{[v_m]\}_{m \in \{1,...M\}} \in \mathbb{R}^d$ is the learnable vectors that substitute the context tokens in the prompt, $M$ is the number of the tokens, and [CLASS] indicates the class names.

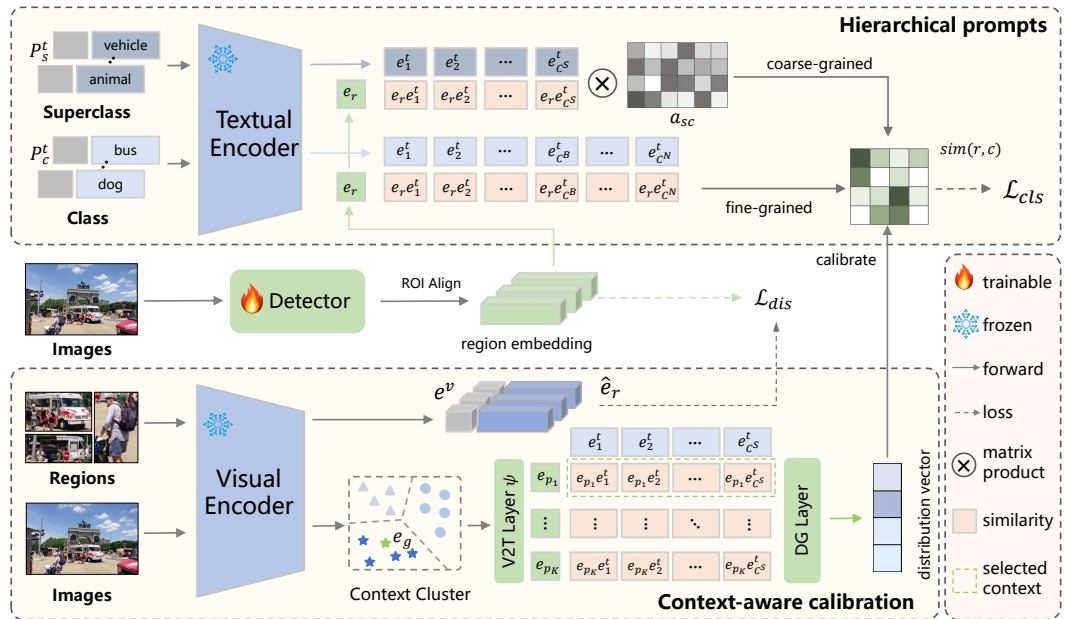

Figure 2: Overall framework of the **Hi**erarchical prompts with **C**ontext-**A**ware calibration (HiCA) for open-vocabulary object detection.

Given an input image $I \in \mathbb{R}^{H \times W \times 3}$, it is processed through the backbone to extract image features. Then the RPN generates a set of proposals $R \in \mathbb{R}^4$ and obtains their region embeddings $\{e_r\}_{r \in R} \in \mathbb{R}^d$ via RoI Align. The similarity for each proposal $r$ and class $c$ is calculated as follows:

$$sim(r, c) = \frac{e_r \cdot e_c^t}{\|e_r\| \cdot \|e_c^t\|}. \tag{1}$$

Current methods focus on directly associating proposal regions with specific categories through prompt tuning. And they often neglect visual context, making it impossible to exploit the association between categories and the context. These limitations significantly constrain the detector's adaptive learning potential when encountering previously unseen novel classes, ultimately restricting its overall generalization ability.

To address the aforementioned issues, we propose **Hi**erarchical prompts with **C**ontext-**A**ware calibration (HiCA) for open-vocabulary object detection. Hierarchical prompting layers learnable prompts based on categories and their coarse-grained descriptions, enabling the model to learn visual region features and align them from coarse-grained prompts to fine-grained prompts. Additionally, we introduce context-aware calibration that captures the real-world distribution of categories within their environmental context. The overall architecture of the proposed method is shown in Figure 2.

### 3.3 HIERARCHICAL PROMPTS FOR MULTI-MODAL KNOWLEDGE DISTILLATION

Existing OVD methods generally train prompts to directly associate proposal regions to predefined object categories, creating a reliance on a fixed set of base categories and restricting their adaptability to novel ones. To overcome this limitation, we propose hierarchical prompts for the multi-modal knowledge distillation method, which learns the mapping between regions and categories in a graduated manner, from shallow to deep by incorporating coarse-grained textual knowledge between visual regions and category descriptions. Thereby improving generalization to unseen novel categories.

This novel approach structures prompt construction into two hierarchical levels: coarse-grained shallow prompts that capture the superclass-to-region relationship, and fine-grained prompts that map regions to specific categories. Superclasses encapsulate broader, higher-level semantic information, facilitating better generalization to novel categories. By progressively structuring trainable prompts

across these hierarchical stages, the model enhances its capacity for generalization and reasoning, resulting in more robust performance on previously unseen novel classes.

Specifically, to achieve more generalized alignment between the target visual features and the text space, we deploy coarse-grained shallow prompts grounded in formulation construction of these coarse-grained prompts $P_s^t = \{[v_1^s], [v_2^s], ...[v_{M_1}^s], [\text{SUPERCLASS}]\}$ follows the form of CoOp (Zhou et al., 2022a), where $\{v_m^s \in \mathbb{R}^d\}_{m=1}^{M_1}$ are learnable vectors used to replace the template context tokens, $M_1$ is the number of the tokens, and [SUPERCLASS] refers to the name of superclasses. We use $P_s^t$ and text encoder $E_t$ to compute the coarse-grained embedding $e_s^t = E_t(P_s^t)$. However, leveraging region embedding $e_r$ and $\{e_s^t\}_{s=1}^{|C^S|} \in \mathbb{R}^{C^S \times d}$ can only obtain the relationship between regions and superclasses. It is necessary to exploit the subordination between superclasses and categories to further obtain the mapping between regions and categories. Hence we construct a subordinate matrix $A = \{a_{ij}\} \in \mathbb{R}^{C^S \times C}$. The element $a_{ij}$ represents the relationship between category $j$ and superclass $i$. If category $j$ belongs to superclass $i$ $a_{ij} = 1$, and $a_{ij} = 0$ otherwise. Thus, we compute the coarse-grained logits between region $r$ and category $c$ belonging to superclass $s$ as follows:

$$sim_{coarse} = \frac{e_r \cdot e_s^t}{\|e_r\| \cdot \|e_s^t\|} \otimes a_{sc}, \tag{2}$$

where $\otimes$ represents matrix multiplication. Similar to coarse-grained logits, we construct fine-grained prompts of categories $P_c^t = \{[v_1^c], [v_2^c], ..., [v_{M_2}^c], [\text{CLASS}]\}$ and extracted category embeddings $\{e_c^t\}_{c=1}^{|C|} \in \mathbb{R}^{C \times d}$ of $P_c^t$ via text encoder. In contrast, fine-grained logits can be directly computed by the similarity between the region embedding $e_r$ and $e_c^t$.

The coarse-grained logits contain rich superior-level semantic information, which can effectively alleviate the overfitting of the base classes and improve the generalization ability of the model to novel classes. While fine-grained logits can improve base class detection performance through labeled data. The combination of the two improves the performance of the base class while maintaining the generalization ability to novel classes. Therefore, we adopt a balance parameter $\lambda$ to reconcile and make the method achieve the best performance:

$$sim(r, c) = \lambda sim_{coarse} + (1 - \lambda) \frac{e_r \cdot e_c^t}{\|e_r\| \cdot \|e_c^t\|}. \tag{3}$$

In the knowledge distillation framework, text prompts are usually used to improve the classification ability of open-vocabulary detectors, while visual prompts are used to improve the performance in extracting regional features. Similar to text prompts, visual prompts are usually fused with visual regions at the input of the VLMs and then encoded by a visual encoder. However, due to the large number of regions generated during the detection process, using visual prompts at the input consumes enormous computation. To reduce the cost, we directly apply learnable visual prompts $e^v$ to the extracted regional embeddings. It preserves the effect without any additional manipulation of the original proposals.:

$$\mathcal{L}_{vp}^O = L1(e_r, \hat{e}_r \oplus e^v). \tag{4}$$

## 3.4 Context-aware Calibration

When transferring features from the VLM model to the detector network, the emphasis is often placed on cropped region features, which tend to neglect the surrounding environmental background. However, the contextual information in the background can enhance the model's ability to generalize to novel classes by expanding high-level visual semantic understanding. To harness this potential, we propose a context-aware probability matrix that establishes connections between context and categories, calibrating the logits produced by the detector classifier. It embodies the probability of a category's occurrence in a given context, utilizing unsupervised contextual clustering alongside textual cues that pertain to both the category and its superclass.

To effectively calculate the context-aware matrix $M^{ca} \in \mathbb{R}^{K \times C^S}$, it is essential to learn the distribution of superclasses in the environmental context, where $K$ represents the number of contextual scenarios. Since contextual information is not contained in the dataset, we need to obtain context embeddings in an unsupervised condition and subsequently refine the matrix. Specifically, we employ the visual encoder $E_v$ of pre-trained VLMs to extract the global features $e_g \in \mathbb{R}^d$ of

the input images $I$ and adaptively cluster the global features to obtain visual context embeddings $e_p = \text{K-means}(E_v(I))$. Moreover, to enhance the detector's ability to capture distribution patterns, we introduce a Distribution Generation (DG) Layer to establish the relationship between context, superclass, and category by learning the distribution of superclasses in the context. Since we can only get the name of the novel classes, we calculate the similarity with the visual context embeddings $e_p \in \mathbb{R}^{K \times d}$ using only the superclass embeddings $\{e_s^t\} \in \mathbb{R}^{C^S \times d}$, where $d$ is the feature dimension:

$$M^{ca} = DG(\psi(e_p) \otimes e_s^{t\top}), \tag{5}$$

the $\psi$ is a fully connected network used to map the context embeddings $e_p$ to the text feature space, to better compute the similarity between $e_p$ and $e_c^t$. We use a multi-layer perception (MLP) to implement the DG layer.

In this way, we obtain the context-aware matrix describing the association between the superclasses and the context. Since the superclass has labeling information, the true distribution frequency of the superclass in contexts can be obtained through truth annotations. We use the ground truth distribution frequency matrix $M^{dst}$ of the base class in the contexts to supervise the generation of the $M^{ca}$ so that the DG layer can learn the ability to map the similarity to the distribution probability:

$$\mathcal{L}_{ca} = L1(M^{ca}, M^{dst}). \tag{6}$$

As long as the similarity between the contexts and the superclass can be obtained, the distribution probability between them can be captured. With $M^{ca}$ and subordinate matrix $A = \{a_{ij}\} \in \mathbb{R}^{C^S \times C}$, we can indirectly calculate the distribution probability $\{\hat{M}^{ca} = M^{ca} \otimes A\} \in \mathbb{R}^{K \times C}$ of novel and base classes in the context and to calibrate the final detection results:

$$P(r, c) = \frac{\exp(sim(r, c) \odot \{\hat{M}_k^{ca}\}_{k \in K})}{\sum\limits_{c' \in C} \exp(sim(r, c') \odot \{\hat{M}_k^{ca}\}_{k \in K})}. \tag{7}$$

The $\odot$ denotes the Hadamard product.

## 3.5 TRAINING AND INFERENCE

**Training** During training, we use the distillation framework of OADP to preserve not only the loss function of Faster R-CNN $\mathcal{L}_{frcnn} = \mathcal{L}_{rpn} + \mathcal{L}_{cls} + \mathcal{L}_{reg}$, but also the global $\mathcal{L}^G$, block $\mathcal{L}^B$, and object $\mathcal{L}^O$ distillation losses. At the same time, based on the original distillation loss, we optimize the object distillation loss $\mathcal{L}_{vp}^O$ by adding learnable visual prompts to improve the distillation efficiency of the knowledge in the object regions. After computing the context-aware matrix $\hat{M}_{ca} \in \mathbb{R}^{K \times C}$, we select the context-aware vector $\{\hat{M}_k^{ca}\}_{k \in K}$ based on the context to which the current image belongs and calibrate the logits obtained from the hierarchical prompts. When calculating the visual context embedding using clustering, we save the global features of each image through a queue and update the visual context embedding in a fixed number of rounds divisible by one thousand.

**Inference** During inference, we use the hierarchical prompts saved during training to calculate the classification score of the detector and do not use the test data to update novel categories' hierarchical prompts. Meanwhile, when leveraging the context-aware matrix in the inference phase, we also select the context-aware vector to calibrate the region-category logits calculated by the knowledge distillation framework according to the current context.

**Discussion** Both our proposed hierarchical prompts and context-aware calibration are independent of the knowledge distillation framework and are plug-and-play flexible modules. Hierarchical prompts only adjust category and superclass prompts according to different datasets and can directly replace prompts in various methods. When the subordination matrix $A = \{a_{ij}\}$ is difficult to generate with annotations, the superclass-category similarity can be used instead. Context-aware calibration uses context embeddings and text prompts independently to learn the distribution of categories and directly acts on region-category logits.

# 4 EXPERIMENT

## 4.1 DATASETS

Building on various open-vocabulary detection methodologies (Gu et al., 2022; Wang et al., 2023; Wu et al., 2023a), we adopt two widely-used open-vocabulary object detection datasets, OV-COCO and OV-LVIS, to thoroughly assess the performance of our approach.

**OV-COCO** In accordance with the setup of (Zareian et al., 2021), the categories in the COCO dataset are re-classified into 48 base categories and 17 novel categories. There are three main metrics to evaluate the performance of the model, $mAP_N$, $mAP_B$, and $mAP_{50}$, where $mAP_N$ represents the mAP of the novel categories with an IoU threshold of 0.5, while $mAP_B$ and $mAP_{50}$ represent the base categories and all categories, respectively.

**OV-LVIS** The dataset originally contained three broad classes: common, frequent, and rare. Following the setting of (Gu et al., 2022), the original rare class has been further subdivided into novel classes, while the common and frequent classes are jointly divided into base classes. Consequently, the OV-LVIS package now includes 337 novel classes and 866 base classes. For this dataset, we use the same metrics names as $AP_r$, $AP_c$, $AP_f$, and $AP$.

## 4.2 IMPLEMENTATION DETAILS

We train our model using 8 V-100 GPUs with a total batch size of 16. Adhering to the implementation details of (Wang et al., 2023), we use SGD as the optimizer with an initial learning rate of 0.02, momentum of 0.9, and weight decay of 0.0001. We use ViT-B/32 CLIP as a teacher model, and its text and visual encoders are used to generate multimodal prompts. The student model is based on the classic Faster RCNN, initializing its ResNet-50 backbone with SoCo. We trained on OV-COCO for a total of 40,000 iterations and reduced the learning rate at 30,000 iters. For OV-LVIS, We use $2\times$ (24 epochs) training schedule, and the learning rate is divided by 10 at the 16th and 22nd epochs.

## 4.3 COMPARISONS WITH STATE-OF-THE-ARTS

Table 1: Comparison results with other state-of-the-art methods on OV-COCO dataset. Methods with the symbol "†" indicate the reproduction result under the same conditions as the proposed method. "T(cat)" denotes using template prompts filled with category names, while "H" denotes hierarchical prompts, and sup is coarse-grained superclass descriptions. "-" indicates the method does not utilize any prompts.

| Methods | Detector | Prompts | $mAP_{50}$ | $mAP_B$ | $mAP_N$ |
|---------|----------|---------|------|------|------|
| ZSD-YOLO(Xie & Zheng, 2022) | YOLOv5x | - | 19.0 | 31.7 | 13.6 |
| HierKD(Ma et al., 2022) | ATSS | T(cat) | 43.2 | 51.3 | 20.3 |
| PB-OVD(Gao et al., 2022) | MRCNN | T(cat) | 42.1 | 46.1 | 30.8 |
| F-VLM(Kuo et al., 2023) | MRCNN | T(cat) | 39.6 | - | 28.0 |
| OVR-CNN(Zareian et al., 2021) | FRCNN | - | 39.9 | 46.0 | 22.8 |
| LocOv(Bravo et al., 2022) | FRCNN | - | 45.7 | 51.3 | 28.6 |
| VLDet(Lin et al., 2023) | FRCNN | T(cat) | 45.8 | 50.6 | 32.0 |
| XPM(Huynh et al., 2022) | FRCNN | - | 41.2 | 46.3 | 27.0 |
| Detic(Zhou et al., 2022b) | FRCNN | T(cat) | 45.0 | 47.1 | 27.8 |
| BARON(Wu et al., 2023a) | FRCNN | T(cat) | 49.1 | 54.8 | 33.1 |
| CORA(Wu et al., 2023b) | D-DETR | T(cat) | 35.4 | 35.5 | 35.1 |
| RALF(Kim et al., 2024) | FRCNN | T(cat) | 49.0 | 54.5 | 33.4 |
| OADP(Wang et al., 2023) | FRCNN | T(cat) | 47.2 | 53.3 | 30.0 |
| OADP$^\dagger$ | FRCNN | T(cat) | 46.0 | 51.7 | 29.9 |
| OADP + HiCA(Ours) | FRCNN | H(cat+sup) | 50.4 | 57.2 | 31.2 |
| BARON$^\dagger$ | FRCNN | L(cat) | 48.9 | 54.6 | 32.9 |
| BARON + HiCA(Ours) | FRCNN | H(cat+sup) | **53.6** | **59.8** | **36.0** |

**Results on OV-COCO** We compare the results of the state-of-the-art (SOTA) with our proposed method. The experimental results on the OV-COCO dataset are presented in Table 1. By adopting

hierarchical prompts and context-aware calibration, we effectively improve the performance on the base and novel categories compared with the baseline OADP and BARON replicated under equivalent experimental conditions. With the OADP baseline we achieved a performance of 31.2% mAP on the novel categories and a result of 57.2% on the base categories. We surpass the OADP[†] by 1.3% and 5.5% on novel and base classes respectively. In the case of BARON as the baseline, we further obtain the best performance of 36.0% $mAP_N$, which outperform BARON[†] by 3.1%. It proves that our proposed HiCA can greatly improve the performance of the base classes while improving the generalization ability of the detector to novel classes. HiCA did not drastically lose the balance of the open-vocabulary detector to improve performance in novel classes. Compared with other methods, CORA has the most extreme situation. In the case of having the highest novel classes mAP, the performance on the base classes is greatly reduced, resulting in its overall detection performance being far lower than most open vocabulary detection methods.

**Results on OV-LVIS**  We compare the results of the SOTA with our proposed method. The experimental results on the OV-LVIS dataset are presented in Table 2. By adopting hierarchical prompts and context-aware calibration, we surpass the OADP[†] by 4.6%, 5.7%, 5.6%, and 1.9% on AP, $AP_c$, $AP_f$, $AP_r$ respectively. And HiCA achieves the best performance of 24.3% in $AP_r$ when BARON is used as the baseline, which is 1.5% higher than BARON[†].

Table 2: Comparison results with other state-of-the-art methods on OV-LVIS dataset. Methods with the symbol "†" indicate the method result that has been reproduced. "T(cat)" denotes using template prompts filled with category names, while "H" denotes hierarchical prompts, and "sup" is coarse-grained superclass descriptions.

| Methods | Detector | Prompts | AP | $AP_c$ | $AP_f$ | $AP_r$ |
|---|---|---|---|---|---|---|
| Vild-ens(Gu et al., 2022) | MRCNN | T(cat) | 27.8 | 26.5 | 34.2 | 16.7 |
| DetPro(Du et al., 2022) | MRCNN | L(cat) | 28.4 | 27.8 | 32.4 | 20.8 |
| Detic(Zhou et al., 2022b) | MRCNN | T(cat) | 26.8 | 26.3 | 31.6 | 17.8 |
| PromptDet(Feng et al., 2022) | MRCNN | L(cat) | 25.3 | 23.3 | 29.3 | 21.4 |
| CondHead(Wang, 2023) | MRCNN | T(cat) | 29.7 | 28.6 | 35.2 | 19.9 |
| F-VLM(Kuo et al., 2023) | MRCNN | T(cat) | 24.2 | - | - | 18.6 |
| BARON(Wu et al., 2023a) | FRCNN | L(cat) | 29.5 | 29.3 | 32.5 | 23.2 |
| RALF(Kim et al., 2024) | FRCNN | T(cat) | 26.6 | 26.2 | 29.1 | 21.9 |
| OADP(Wang et al., 2023) | FRCNN | T(cat) | 28.7 | 28.4 | 32.0 | 21.9 |
| OADP[†] | FRCNN | T(cat) | 27.8 | 27.6 | 32.1 | 18.9 |
| OADP + HiCA(Ours) | FRCNN | H(cat+sup) | **32.4** | 33.2 | **37.6** | 20.8 |
| BARON[†] | FRCNN | L(cat) | 29.1 | 28.9 | 31.9 | 22.8 |
| BARON + HiCA(Ours) | FRCNN | H(cat+sup) | 32.3 | **34.1** | 37.0 | **24.3** |

## 4.4 Ablation Study

We conduct ablation experiments on the OV-COCO dataset to demonstrate the effectiveness of our proposed method. The baseline is the OADP that we reproduce under the same experimental conditions. HP indicates that hierarchical prompts and learnable visual prompts are used, and CA indicates context-aware calibration.

Table 3: Ablation study of hierarchical prompts with context-aware calibration on OV-COCO dataset.

| Method | $mAP_N$ | $mAP_B$ | $mAP_{50}$ |
|---|---|---|---|
| baseline | 29.9 | 51.7 | 46.0 |
| baseline + HP | 30.0 | **57.5** | 50.3 |
| baseline + HP + CA | **31.2** | 57.2 | **50.4** |

**Hierarchical prompts based multi-modal knowledge distillation**  The role of Hierarchical prompts is to improve the detection performance of the base class while ensuring the generalization ability of the novel class and maintaining the balance of the overall classes. As shown in Table 3, the baseline with hierarchical prompts achieves a significant boost of 5.8% on the base class, while keeping the mAP of the novel class slightly higher than the baseline.

**Context-aware calibration**  Context-aware calibration corrects the detector according to the current context after it obtains the region-category score. The detector learns the distribution of various categories in different contexts by introducing stable information about the features of the context environment. Table 3 shows that with context-aware calibration, the overall open-vocabulary detection performance is improved, but the mAP of the base class is slightly decreased. It indicates that in the same context, the detector is inclined to detect objects with similar appearance as novel classes, which alleviates the preference of the detector for the base class through training and improves the generalization performance of the model for novel classes.

**Balance parameter**  We set a balance parameter $\lambda$ to adjust the proportion of coarse-grained and fine-grained embeddings when constructing hierarchical prompts. We designed two ways to insert the parameters. One is first to calculate the coarse-grained and fine-grained logits separately, and then use $\lambda$ to adjust the final region-category logits, as shown in Figure 3 (a). The other is to use $\lambda$ to fuse the coarse-grained and fine-grained prompts embeddings, and then calculate the logits with the fused embeddings as shown in Figure 3 (b). As shown in Figure 3, both methods have the same trend. The mAP of the detector on the novel and the base class steadily improves with the increase of the proportion of coarse-grained knowledge while the parameter is less than $0.7$. The smaller the balance parameter, the closer the hierarchical prompts are to the traditional learnable prompt, which does not perform well with random initialization. When the value of $\lambda$ exceeds $0.7$, the prompts embedding from the same superclass are too close in the feature space, which makes it difficult to distinguish the novel class from the base ones.

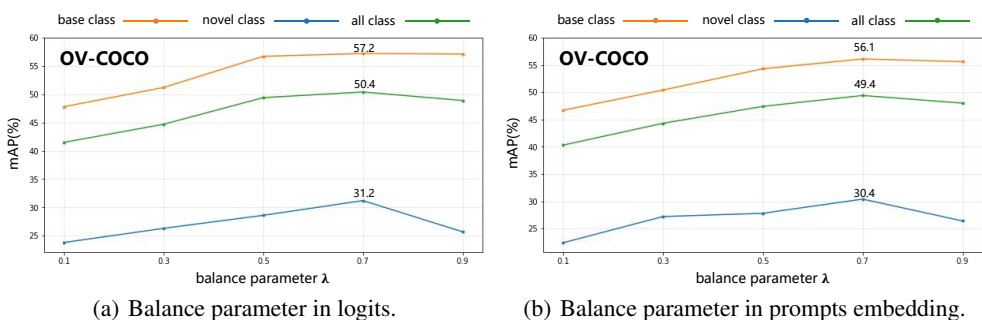

(a) Balance parameter in logits.  (b) Balance parameter in prompts embedding.

Figure 3: Ablation study of balance parameter $\lambda$ on OV-COCO dataset.

**Different prompts type**  During the construction of hierarchical prompts, the experimental effects of different types of prompts are studied based on the knowledge distillation framework. Baseline uses a fixed template to construct category prompts. As shown in Table 4, we initially used learnable prompts to replace the original ones. However, since we only used random initialization, we got poor performance on both the novel and base classes. To further improve the performance of the open-vocabulary detector, we try to construct multi-modal prompts. The results of multi-modal prompts in Table 4 prove that the introduction of learnable visual prompts improves the mAP of the novel and base class by $1.2\%$ and $0.4\%$ respectively compared with the method using only learnable text prompts. Nevertheless, the performance improvement of multi-modal prompts is very limited, hence we design hierarchical prompts that use both coarse-grained and fine-grained information. In the case of only using textual hierarchical prompts, the method has a large improvement of $5.9\%$ compared with baseline on the base class, while the mAP of the novel class only decreases by $1.6\%$, and the overall detection performance is increased by $3.9\%$. It indicates that hierarchical prompts can improve the performance of base classes while maintaining generalization to novel classes. Since multi-modal prompts are proved to have a superior performance by experiments, we add visual prompts to the framework leveraging hierarchical prompts. We end up with a mAP of $30.0\%$ and $57.5\%$ on the novel and base classes, respectively.

Table 4: Ablation study of different prompts on OV-COCO dataset.

| Prompts | Templet | Learnable | $mAP_N$ | $mAP_B$ | $mAP_{50}$ |
|---|---|---|---|---|---|
| Text | ✓ | | 29.9 | 51.7 | 46.0 |
| Text | | ✓ | 23.2 | 46.7 | 40.6 |
| Multi-modal | | ✓ | 24.4 | 47.1 | 41.1 |
| Hierarchical only | | ✓ | 28.3 | **57.6** | 49.9 |
| Hierarchical + Multi-modal | | ✓ | **30.0** | 57.5 | **50.3** |

## 5 VISUALIZATION

We visualize the projection of commonly used prompts and hierarchical prompts in the feature space. As shown in figure 4 (a), when using only the region-category fine-grained prompts, the categories belonging to different superclasses become entangled in the feature space. Although categories from the same superclass tend to be close to each other, categories between different superclasses cannot be clearly distinguished from each other either. Figure 4 (b) illustrates the projection of hierarchical prompts in the feature space. It uses category fine-grained prompts to reduce the distance between categories with similar appearance and leverages coarse-grained prompts to widen the distance between categories belonging to different superclasses. In this way, the false detection of objects with similar appearance but different superclasses can be reduced, improving the detection performance of base classes. It can also increase the detection of objects that are clearly different from the background, maintaining the generalization ability of novel classes.

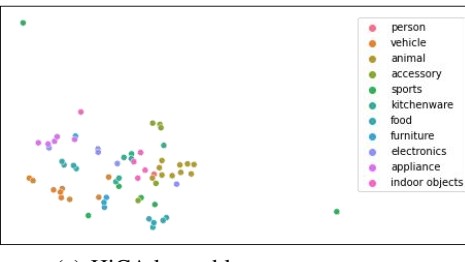 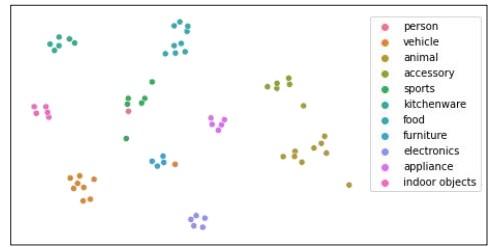

(a) HiCA learnable category prompts.      (b) HiCA learnable hierarchical prompts.

Figure 4: Visualization of the projection of prompts.

## 6 CONCLUSION

In this paper, we propose a **Hi**erarchical prompts with **C**ontext-**A**ware calibration (HiCA) for open-vocabulary object detection to enhance the ability of knowledge distillation framework to transfer high-level semantic knowledge. The core idea of HiCA is to make full use of superior-level semantic information in vision and language and maintain the generalization ability to novel classes while improving the performance of the open vocabulary detector. The hierarchical prompts integrate coarse-grained superclass knowledge as an intermediary step, thereby transforming the region-category into a two-stage process. It ensures that coarse-grained knowledge mitigates potential biases towards base classes during training. Context-aware calibration revises detector results by learning category distribution through environmental context knowledge. We conduct sufficient comparison and ablation experiments to demonstrate the superior performance of our proposed method.

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

## A    APPENDIX

**Visualization for visual-text similarity matrix**    We use the similarity matrix to analyze the discriminative ability of the hierarchical prompts. Figure 5 shows the similarity matrix between the hierarchical prompts embedding and the visual prototype of the category (48 base classes and 6 novel classes). Ideally, the matrix should have light colors on the diagonal (high similarity) and dark colors on the off-diagonal (low similarity). However, the matrix in Figure 5 is not maximal in the diagonal of the novel category (classes 48 to 53), which leads to a limited improvement in the detection performance of the novel class when only using hierarchical prompts. Therefore, context-aware calibration is needed to correct this similarity matrix. Although the context is related to the input and cannot be directly applied to schemas calculated using category prototypes, ablation studies show that context-aware calibration improves HiCA's performance by another 1.2% on novel classes, proving that it can effectively calibrate results with biased similarities.

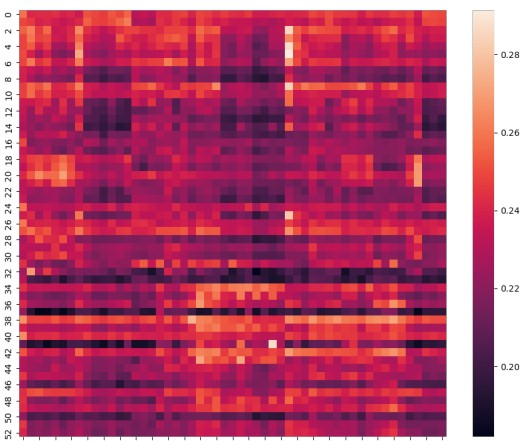

Figure 5: Visualization for visual-text similarity matrix with hierarchical prompts.

**Quantitative analysis of hierarchical prompts**  We analyzed the discriminative power of hierarchical prompts for categories with similar appearances using a similarity matrix. We intercept some representative categories for analysis. Figure 6 (a) shows the similarity matrix of the visual features between different categories, which is obtained by the prototype of each category. The lighter the color, the more similar the appearance between categories. When text embedding is used to classify visual features, the optimal form of the visual-text similarity matrix should be light colors on the diagonal (high similarity) and dark colors on the off-diagonal (low similarity). Figure 6 (b) shows the result of the subtraction of the similarity matrix calculated using hierarchical prompts and single text prompts. The darker in off-diagonal position, the more effective the hierarchical prompt is (the gap between different categories of text and visual features is larger). For example, the similarity between categories 1 to 10 in the upper left corner is high, and the hierarchical prompts effectively improve the discrimination ability in this region, which proves its ability to distinguish categories with high similarity.

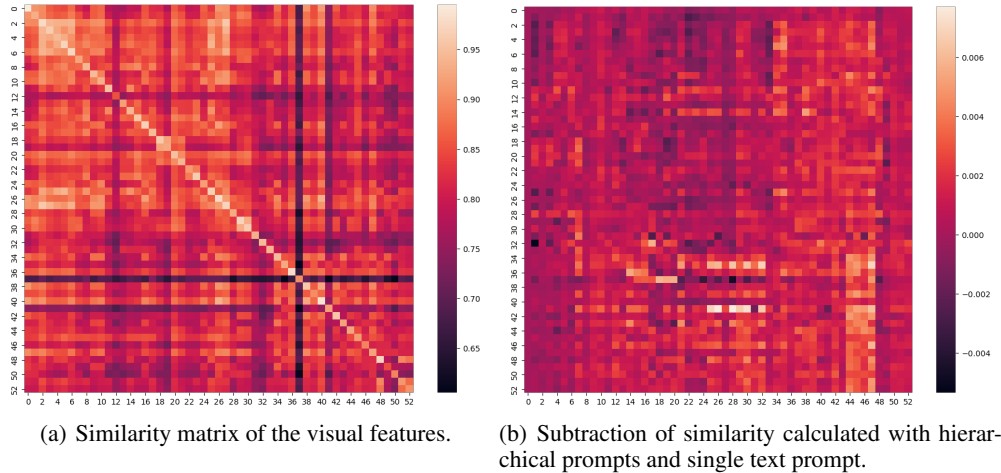

(a) Similarity matrix of the visual features.  (b) Subtraction of similarity calculated with hierarchical prompts and single text prompt.

Figure 6: Quantitative analysis for hierarchical prompts.

**Detailed analysis of context-aware calibration**  As the results shown in Table 5, the performance of the model will decrease if the number of unsupervised context clusters is too large or too small. An increase in the cluster center of the context represents a further subdivision of the environment and is likely to result in more similar context embedding. This can lead to confusion when calculating the distribution matrix. However, if the number of context clusters is too small, some environments will

be mixed and the distribution matrix will not be effective. The purpose of the DG layer is to map the context-superclass similarity matrix into a distribution matrix. A single fully connected layer for the DG layer cannot learn an effective mapping relationship, and too deep MLP may learn some bias in the training process. These reasons will lead to a degradation in performance.

Table 5: Ablation study of context clustering and the DG layer. "Number" represents the number of centers of the context clustering. "Depth" denotes the MLP depth of the DG layer.

| Number | Depth | $mAP_N$ | $mAP_B$ | $mAP_{50}$ |
|---|---|---|---|---|
| 8 | 1 | 29.3 | 54.4 | 47.8 |
| 8 | 2 | **31.2** | **57.2** | **50.4** |
| 8 | 3 | 27.6 | 53.7 | 46.9 |
| 6 | 1 | 30.4 | 54.5 | 48.2 |
| 10 | 1 | 28.5 | 55.4 | 48.3 |

