# OpenReview forum: "Hierarchical Prompts with Context-aware Calibration for Open-Vocabulary Object Detection"
_ICLR.cc/2025/Conference — Submitted to ICLR 2025_

### Official Review · Reviewer_EcPQ · 2024-11-01

**Soundness:** 3
**Presentation:** 3
**Contribution:** 2
**Rating:** 5
**Confidence:** 4

**Summary:**

This paper introduces a novel approach, Hierarchical Prompts with Context-Aware Calibration (HiCA), to enhance Open Vocabulary Object Detection (OVD). HiCA improves generalization by utilizing hierarchical prompts that map object regions through both coarse- and fine-grained knowledge, capturing shared information across base and novel classes. Additionally, context-aware calibration refines detection by linking contextual information with object categories, reducing background interference.

**Strengths:**

1. Clear Motivation: The motivation behind the proposed method is intuitive and well-aligned with the goals of Open Vocabulary Detection (OVD), making it easy to follow and logically sound.
2. Comprehensible Diagrams: The diagrams used in the paper are clear and well-illustrated, which aid in understanding the overall architecture and key components of the proposed framework.
3. Extensive Experiments: The paper provides comprehensive experimental evaluations, covering various settings and comparisons, which helps to validate the effectiveness of the proposed method.

**Weaknesses:**

1. Limited Improvement for Novel Classes: Despite the emphasis on enhancing detection for novel classes in OVD, the improvement shown in Tables 1 and 4 is relatively minor. For instance, in the ablation study, adding hierarchical prompts (HP) increases performance on novel classes by only 0.01, which raises questions about the overall impact of this component for new class detection. It is suggested to add per-class performance breakdowns or error analysis to explain this phenomenon.
2. Lack of Quantitative Analysis: The paper claims that the introduction of superclasses improves differentiation between visually similar classes. However, the paper lacks quantitative analysis or clear evidence to support this claim. It would benefit from an in-depth exploration of cases (e.g., a confusion matrix analysis or a comparison of specific cases) where superclasses improve detection performance and an explanation of the mechanisms by which HiCA differentiates similar classes more effectively.
3. Concern of Prompt Design: According to Table 4, most modifications do not outperform simpler, template-based methods for novel classes. This raises concerns about the practicality of the proposed approach for novel class detection. Considering the limited improvement, a more detailed discussion of the scenarios where the method might be preferable would be helpful.

**Questions:**

1. Impact of Knowledge Granularity: Does the granularity level of knowledge (i.e., fine-grained versus coarse-grained superclass distinctions) affect the model's performance? More insights on the optimal level of superclass granularity would be valuable.
2. Generalizability of Handcrafted Superclasses: Hand-designed superclasses could impact the method's adaptability in real-world applications. If a new superclass emerges, does the model need fine-tuning again? Additionally, how does HiCA handle ambiguous cases that do not fit well into existing superclasses? It is suggested that the paper could provide examples to illustrate how HiCA would work in such situations or how HiCA could be adjusted.

---

> ### Author Response · Authors · 2024-11-24
>
> Dear reviewer, Thank you for your thorough review and valuable comments. Below are our responses to your concerns and questions.
>
> 1. Limited Improvement for Novel Classes
>
>     Open vocabulary detection methods should indeed pay more attention to the performance in novel classes. In the previous experiments, although the improvement of HiCA on novel classes is limited, it still can steadily improve the detection ability for novel categories. By analyzing it down to the categories, we find the performance degradation on individual categories limits the average performance of HiCA on the overall novel class. e.g. that HiCA achieves 2.3% and 2.8% performance improvement for "bus" and "dog" categories, and 0.5% degradation for "elephant". Meanwhile, the performance of HiCA is also limited by the selection of baseline. Since the OADP itself is not comparable to SOTA, HiCA has not been able to achieve results close to SOTA in the novel class. To this end, HiCA as a plug-and-play method should further explore the impact of baseline. BARON is used as the new baseline to compare with other OVD methods using prompts on OV-COCO and OV-LVIS (Table 1 and Table 2 in the paper). The best performance of 36.0% $\mathrm{mAP}_N$ and 24.3% $\mathrm{AP}_r$ is obtained on novel classes, respectively. Experiments show that HiCA can perform better when combined with a suitable high-performance baseline.
>
>    |||||||
>    |-|-|-|-|-|-|
>    |Methods|Detector|Prompts|$\mathrm{mAP}_{50}$|$\mathrm{mAP}_B$|$\mathrm{mAP}_N$|
>    |$\mathrm{BARON}$|FRCNN|L(cat)|48.9|54.6|32.9|
>    |$\mathrm{BARON + HiCA(Ours)}$|FRCNN|H(cat+sup)|**53.6**|**59.8**|**36.0**|
>
>
>    ||||||||
>    |-|-|-|-|-|-|-|
>    |Methods|Detector|Prompts|$\mathrm{AP}$|$\mathrm{AP}_c$|$\mathrm{AP}_f$|$\mathrm{AP}_r$|
>    |$\mathrm{BARON}$|FRCNN|L(cat)|29.1|28.9 |31.9|22.8 |
>    |$\mathrm{BARON + HiCA(Ours)}$|FRCNN|H(cat+sup)|32.3|**34.1**|37.0|**24.3**|
>
> 2. Lack of Quantitative Analysis
>
>     Following the suggestions made by the reviewers, we analyzed the discriminative ability of hierarchical prompts for categories with similar appearances. We intercept some representative categories for analysis. Figure 6 (a) in the appendix shows the similarity matrix of the visual features. The lighter the color, the more similar the appearance between categories. Figure 6 (b) shows the result of the subtraction of the similarity matrix calculated using hierarchical prompts and single text prompts. The darker in off-diagonal position, the more effective the hierarchical prompt is (the gap between different categories of text and visual features is larger). For example, the similarity between categories 1 to 10 in the upper left corner is high, and the hierarchical prompt effectively improves the discrimination ability in this region, which proves its ability to distinguish categories with high similarity.
> 3. Concern of Prompt Design
>
>     The ablation studies in Table 4 are implemented based on the baseline OADP, while OADP itself uses a template-based prompt. This may require more supervision and initialization information to adjust and optimize the adaptation of the learnable prompts to the model. If the baseline is originally using a learnable prompt, it can not only naturally use the originally effective setting, but also better fit the model. For example, when BARON is used as the baseline, HiCA can improve the performance of the novel class by 3.1%. It can be concluded that HiCA works better when combined with a high-performance baseline, which already uses a learnable prompt.
>
> **Questions:**
>
> 1. Impact of Knowledge Granularity
>
>    The first thing to note is that different granularity levels of knowledge can affect model performance. Our view of the superclass level is that it is sufficient as long as the superclass covers all categories. For example, COCO and LVIS use the same images, but LVIS has a more fine-grained category division, so we consider COCO and LVIS to share a common superclass. Experiments show that using the same superclass set on the two datasets can improve the model performance on both datasets. If we need to verify the specific level to which the superclass has the optimal performance, we need to reconstruct the scientific superclass level for experiments. It can be used in our future research for more in-depth exploration.
> 2. Generalizability of Handcrafted Superclasses
>
>     The model needs to be fine-tuned for better performance if the novel superclass is not part of the existing superclass. When the superclass is not suitable for the current situation, we can first obtain discriminative category text features through the zero-shot learning ability of HiCA to realize the detection of the category. However, this fails to construct a relationship between the novel and the base class, it may result in a relatively low mAP for the class. We can further add the new superclasses to the existing superclasses list and retrain the model for better performance.

---

> > ### Comment · Reviewer_EcPQ · 2024-11-28
> >
> > I thank the authors for their response by conducting further experiments and analyses. The answers addressed some of my concerns, however, I still think the real improvement of the proposed technique is not significant and have concerns on its practical usage in real-world scenario. I'd suggest the authors to improve their work by exploiting more strong baselines and conduct further in-depth exploration on the prompt design. Therefore, I will maintain my score.

---

### Official Review · Reviewer_ZtTR · 2024-11-02

**Soundness:** 2
**Presentation:** 3
**Contribution:** 2
**Rating:** 5
**Confidence:** 3

**Summary:**

This paper proposes HiCA, a novel approach for open-vocabulary object detection. In HiCA, Hierarchical prompts leverage coarse-grained superclass knowledge to avoid biasing the model towards base classes, and Context-Aware calibration revises detector results by learning category distribution through environmental context knowledge. Experimental results demonstrate that HiCA consistently outperforms state-of-the-art methods in detecting novel objects.

**Strengths:**

1. The paper is well-written and easy to understand.
2. The idea of using Hierarchical prompts to capture the shared knowledge between both base and novel classes is interesting and proven.
3. Experimental results show that the proposed method is effective.

**Weaknesses:**

1. The author claims that Hierarchical prompts can enhance the model's generalization ability for novel classes, but Table 3 shows that its performance improvement for novel classes is small.
2. Although Context-Aware calibration can bring performance improvement, there is a lack of detailed experimental verification of the effectiveness of the motivation.
3. The author claims that Hierarchical prompts and Context-Aware calibration are plug-and-play modules, but no further experimental verification is performed.

**Questions:**

1. In Tables 1 and 2, the performance of the base classes achieves state-of-the-art results, while the novel classes perform poorly. Should we focus more on enhancing the performance of the novel classes?
2. This paper lacks a detailed explanation of how the selected context in Figure 2 is implemented.

---

> ### Author Response · Authors · 2024-11-24
>
> Dear reviewer, Thank you for your thorough review and valuable comments. Below are our responses to your concerns and questions.
>
> 1. Table 3 shows that its performance improvement for novel classes is small.
>
>     Open vocabulary detection methods should indeed pay more attention to the performance in novel classes. In the previous experiments, although the improvement of HiCA on novel classes is limited, it still can steadily improve the detection ability for novel categories. Meanwhile, the performance of HiCA is also limited by the selection of baseline. Since the OADP itself is not comparable to SOTA, HiCA has not been able to achieve results close to SOTA in the novel class. To this end, HiCA as a plug-and-play method should further explore the impact of baseline. BARON is used as the new baseline to compare with other OVD methods using prompts on OV-COCO and OV-LVIS (Table 1 and Table 2 in the paper).  The best performance of 36.0% $\mathrm{mAP}_N$ and 24.3% $\mathrm{AP}_r$ is obtained on novel classes, respectively. Experiments show that HiCA can perform better when combined with a suitable high-performance baseline.
>
>    |||||||
>    |-|-|-|-|-|-|
>    |Methods|Detector|Prompts|$\mathrm{mAP}_{50}$|$\mathrm{mAP}_B$|$\mathrm{mAP}_N$|
>    |$\mathrm{BARON}$|FRCNN|L(cat)|48.9|54.6|32.9|
>    |$\mathrm{BARON + HiCA(Ours)}$|FRCNN|H(cat+sup)|**53.6**|**59.8**|**36.0**|
>
>
>    ||||||||
>    |-|-|-|-|-|-|-|
>    |Methods|Detector|Prompts|$\mathrm{AP}$|$\mathrm{AP}_c$|$\mathrm{AP}_f$|$\mathrm{AP}_r$|
>    |$\mathrm{BARON}$|FRCNN|L(cat)|29.1|28.9 |31.9|22.8 |
>    |$\mathrm{BARON + HiCA(Ours)}$|FRCNN|H(cat+sup)|32.3|**34.1**|37.0|**24.3**|
>
> 2. There is a lack of detailed experimental verification of the effectiveness of the context-aware calibration motivation.
>
>     To further show the details of the context-aware calibration, we add new ablation studies for the context clustering and the DG layers. The new ablation studies results show how context knowledge can be used effectively (Table 5 in the appendix). As shown below, the performance will decrease if the number of context clusters is too large or too small. An increase in the cluster center represents a further subdivision of the context and is likely to result in more similar context embedding. This can lead to confusion when calculating the distribution matrix. However, if the number of context clusters is too small, some environments will be mixed and the distribution matrix will not be effective. For the depth of the DG layer, a single fully connected layer cannot learn an effective mapping relationship, and too deep MLP may learn some bias in the training process. These reasons will lead to a degradation in performance.
>
>    ||||||
>    |-|-|-|-|-|
>    |Number|Depth|$\mathrm{mAP}_{50}$|$\mathrm{mAP}_B$|$\mathrm{mAP}_N$|
>    |8|1|29.3|54.4|47.8|
>    |8|2|**31.2**|**57.2**|**50.4**|
>    |8|3|27.6|53.7|46.9|
>    |6|1|30.4|54.5|48.2|
>    |10|1|28.5|55.4|48.3|
>
> 3. No further experiment verifies the HiCA is a plug-and-play module.
>
>     We also recognize that HiCA as a plug-and-play method should further explore the impact of baseline on model performance. Therefore, we use BARON as a baseline to compare with other OVD methods. Subject to word count limitation, experimental results and analysis can be found in the answer to weakness 1.
>
> **Questions:**
>
> 1. In Tables 1 and 2, the performance of the base classes achieves state-of-the-art results, while the novel classes perform poorly. Should we focus more on enhancing the performance of the novel classes?
>
>     Yes, as open vocabulary detection methods should indeed pay more attention to the detection performance in novel classes. Therefore, we analyzed the problems in the previous experiments and found that the choice of baseline would have a great impact on the performance of HiCA. In new supplementary experiments, we demonstrate that HiCA performs better when combined with a suitable high-performance baseline.
> 2. This paper lacks a detailed explanation of how the selected context in Figure 2 is implemented.
>
>     The specific selection process is as follows. Firstly, K-means is used to cluster the overall feature of the input image, and the environment K to which the current image belongs is obtained. Next, The image feature is used to update the cluster center. Then, the context-superclass similarity matrix is calculated by using the updated clustering center and the superclass text embedding. This matrix is mapped to the context-superclass distribution matrix $M^{ca}$ by the DG layer. Using $M^{ca}$ and subordinate matrix $A$, we further calculate the distribution matrix of category $\hat{M}^{ca}$ in the environment. According to the context to which the current image belongs, the *k*-th row vector is extracted from $\hat{M}^{ca}$ as the distribution vector of each category. The vector is element-by-element multiplied by the logits matrix calculated by the classifier to obtain the modified classification score.

---

### Official Review · Reviewer_eWpv · 2024-11-02

**Soundness:** 3
**Presentation:** 3
**Contribution:** 2
**Rating:** 5
**Confidence:** 4

**Summary:**

The paper proposes a novel method, Hierarchical prompts with Context-Aware Calibration (HiCA), for open vocabulary object detection that integrates both linguistic and visual contextual information to improve generalization to novel classes. The approach achieves good results on the existing COCO and LVIS benchmark.

**Strengths:**

1. The method of combining superclass and class to form hierarchical prompts is interesting.
2. The method enhances the performance of the OADP model on both COCO and LVIS datasets.

**Weaknesses:**

1. I was wondering that the focus of the hierarchical prompts proposed by the author may be inconsistent with the main goal of open vocabulary object detection. Specifically, according to references [1, 2, 3], the primary goal of open vocabulary object detection is to enhance the generalization ability to novel classes while maintaining detection performance for base classes.  Although the author mentions in the abstract that “Hierarchical prompts enhance the model’s generalization ability to novel classes,” the ablation study highlights its effectiveness for base classes with a 5.8% boost, while novel classes see only a marginal 0.1% increase over the baseline.
2. The ablation experiments and visualizations primarily focus on verifying the effectiveness of hierarchical prompts, but they lack a detailed analysis of context-aware calibration, particularly concerning the effectiveness of context clustering, and the DG Layer.
3. In Section 3.5, the author mentions the proposed modules are plug-and-play flexible modules. However, the paper only provides experimental results integrated with the OADP method. Could the author provide results on other state-of-the-art methods to demonstrate the plug-and-play effectiveness of these modules?
4. Lack of separate experimental analysis on the effectiveness of learnable visual prompts.

[1] Learning Object-Language Alignments for Open-Vocabulary Object Detection, ICLR 2023
[2] Multi-Modal Classifiers for Open-Vocabulary Object Detection, ICML 2023
[3] Aligning Bag of Regions for Open-Vocabulary Object Detection, CVPR 2023

**Questions:**

1. How is the superclass determined for each category, considering that a category could be associated with multiple levels of superclasses? Furthermore, which level of superclass demonstrates greater effectiveness?
2. Please refer to the weakness section.

---

> ### Author Response · Authors · 2024-11-24
>
> Dear reviewer, Thank you for your thorough review and valuable comments. Below are our responses to your concerns and questions.
>
> 1. The focus of the hierarchical prompts proposed by the author may be inconsistent with the main goal of OVD. The ablation study highlights its effectiveness for base classes with a 5.8% boost, while novel classes see only a marginal 0.1% increase over the baseline.
>
>     Open vocabulary detection methods should indeed pay more attention to the performance in novel classes. In the previous experiments, although the improvement of HiCA on novel classes is limited, it still can steadily improve the detection ability for novel categories. By analyzing it down to the categories, we find the performance degradation on individual categories limits the average performance of HiCA on the overall novel class. Meanwhile, the performance of HiCA is also limited by the selection of baseline. Since the OADP itself is not comparable to SOTA, HiCA has not been able to achieve results close to SOTA in the novel class. To this end, HiCA as a plug-and-play method should further explore the impact of baseline. BARON is used as the new baseline to compare with other OVD methods using prompts on OV-COCO and OV-LVIS (Table 1 and Table 2 in the paper).  The best performance of 36.0% $\mathrm{mAP}_N$ and 24.3% $\mathrm{AP}_r$ is obtained on novel classes, respectively. Experiments show that HiCA can perform better when combined with a suitable high-performance baseline.
>
>    |||||||
>    |-|-|-|-|-|-|
>    |Methods|Detector|Prompts|$\mathrm{mAP}_{50}$|$\mathrm{mAP}_B$|$\mathrm{mAP}_N$|
>    |$\mathrm{BARON}$|FRCNN|L(cat)|48.9|54.6|32.9|
>    |$\mathrm{BARON + HiCA(Ours)}$|FRCNN|H(cat+sup)|**53.6**|**59.8**|**36.0**|
>
>
>    ||||||||
>    |-|-|-|-|-|-|-|
>    |Methods|Detector|Prompts|$\mathrm{AP}$|$\mathrm{AP}_c$|$\mathrm{AP}_f$|$\mathrm{AP}_r$|
>    |$\mathrm{BARON}$|FRCNN|L(cat)|29.1|28.9 |31.9|22.8 |
>    |$\mathrm{BARON + HiCA(Ours)}$|FRCNN|H(cat+sup)|32.3|**34.1**|37.0|**24.3**|
>
> 2. The paper lacks a detailed analysis of context-aware calibration, particularly concerning the effectiveness of context clustering, and the DG Layer.
>
>    We add new ablation studies for the context clustering and the DG layers. The results are shown as follows (Table 5 in the appendix). The performance will decrease if the number of context clusters is too large or too small. An increase in the cluster center represents a further subdivision of the context and is likely to result in more similar context embedding. This can lead to confusion when calculating the distribution matrix. However, if the number of context clusters is too small, some environments will be mixed and the distribution matrix will not be effective. For the depth of the DG layer, a single fully connected layer cannot learn an effective mapping relationship, and too deep MLP may learn some bias in the training process. These reasons will lead to a degradation in performance.
>
>    ||||||
>    |-|-|-|-|-|
>    |Number|Depth|$\mathrm{mAP}_{50}$|$\mathrm{mAP}_B$|$\mathrm{mAP}_N$|
>    |8|1|29.3|54.4|47.8|
>    |8|2|**31.2**|**57.2**|**50.4**|
>    |8|3|27.6|53.7|46.9|
>    |6|1|30.4|54.5|48.2|
>    |10|1|28.5|55.4|48.3|
>
> 3. Could the author provide results on other state-of-the-art methods to demonstrate the plug-and-play effectiveness of these modules?
>
>     We also recognize that HiCA as a plug-and-play method should further explore the impact of baseline on model performance. Therefore, we use BARON as a baseline to compare with other OVD methods. Subject to word count limitation, experimental results and analysis can be found in the answer to weakness 1.
>
> 4. Lack of separate experimental analysis on the effectiveness of learnable visual prompts.
>
>     We provide a separate experimental analysis of learnable visual prompts in Table 4, and we name it "Multi-modal". Since HiCA is a method based on prompt tuning, the classes' text embedding needs to be used as the classifier of the OVD detector, so the learnable visual prompt cannot stand alone.
>
> **Questions:**
>
> 1. How is the superclass determined for each category, considering that a category could be associated with multiple levels of superclasses? Furthermore, which level of superclass demonstrates greater effectiveness?
>
>     The superclass we use is the annotation information provided by the COCO dataset. Our view of the superclass level is that it is sufficient as long as the superclass covers all categories. For example, COCO and LVIS use the same images, but LVIS has a more fine-grained category division, so we consider COCO and LVIS to share a common superclass. Experiments show that using the same superclass set on the two datasets can improve the model performance on both datasets. If we need to verify the specific level to which the superclass has the optimal performance, we need to reconstruct the scientific superclass level for experiments. It can be used in our future research for more in-depth exploration.

---

### Official Review · Reviewer_Cb1c · 2024-11-04

**Soundness:** 3
**Presentation:** 3
**Contribution:** 2
**Rating:** 5
**Confidence:** 5

**Summary:**

This paper proposes two modules to enhance the standard Faster R-CNN-based paradigm for open-vocabulary object detection. The first motivated by the limited semantic representation of category names in textual form, and it calibrates the original textual features with those derived from superclass features. The second one leverages visual context information from images, encode context features by a visual encoder and then used to further refine the text embeddings. Experiments on OV-COCO and OV-LVIS demonstrate a measurable improvement.

**Strengths:**

1. The paper is easy to follow and well organized.  The figures are drawn well.
2.  The two modules are well-motivated.  Utilizing the superclass along with the category intuitively polishes and enriches the textual representation of each class.  The contextual feature also plays a critical role in detection, which is usually neglected but could potentially build the relation graph between novel and base categories and bring more improvements.
3.  The experiments follow the typical setting of open-vocabulary object detection, and the ablation study presents the effectiveness of each module.

**Weaknesses:**

Despite the well-established motivation of this paper, it contains several weaknesses in my consideration.

1. Enriching textual description of the categories beyond a single word name is not a novel idea.   Some previous work like [R1]  has proved the effectiveness of it.   Instead of improving the textual description, the paper involves another module to calibrate the original textual feature for classification.  This enrolls more computation overhead and is less elegant than directly fineturing the existing textual encoder.
2. Compared to methods that utilize enriched information from large language models (LLMs), the use of superclasses from a word tree offers limited enhancement to the textual features, providing only modest complementary information. This limitation is evident in the ablation results shown in Table 3.
3.  Especially, for open-vobulary detection, the critical metric is AP-novel, not AP-base or AP-all, since the core concern is the model’s generalization capability. In the experiments, both modules show only limited improvements in this essential metric.


[R1] Prannay Kaul, Weidi Xie, Andrew Zisserman. Multi-Modal Classifiers for Open-Vocabulary Object Detection. ICML 2023.

**Questions:**

I have no additional questions confusing me but the concerns listed in the weakness section.

---

> ### Author Response · Authors · 2024-11-24
>
> Dear reviewer, Thank you for your thorough review and valuable comments. Below are our responses to your concerns and questions.
>
> 1. Enriching textual categories' descriptions is not a novel idea. Previous work like [R1] has proved its effectiveness. The paper involves another module to calibrate the original textual feature for classification. This enrolls more computation overhead and is less elegant than directly fineturing the existing textual encoder.
>
>     HiCA uses hierarchical prompts to effectively map the shared multimodal knowledge between base and novel classes, thereby enhancing the model's generalization ability to novel classes, unlike the operation of enriching textual category descriptions in [R1]. The calibration module corrects the final classification logic by estimating the distribution of superclass and category in the environmental context, and its additional calculations can be ignored.
>
>     (1) Hierarchical prompts use advanced superclass knowledge to establish associations between base and novel classes, to alleviate the model's preference for base classes. However, [R1] uses LLM to enrich the description of each class and obtain more discriminative text embeddings, which is fundamentally different from our approach of using prompt tuning and cannot achieve the effect of shared knowledge transfer.
>
>     (2) Conversely, using LLM to generate additional text descriptions will increase the input data. Taking [R1] as an example, it generates a total of 12030 category descriptions for OV-LVIS. Our method only uses the class and superclass names provided by the dataset without any additional data.
>
>    (3) Context-aware calibration corrects the final classification logic by estimating the distribution of categories in the context, rather than directly processing the original text features. The computational complexity of the baseline (BARON) and HiCA models is 0.654T FLOPs, indicating that they will not increase computational complexity.
>
> 2. The use of superclasses from a word tree offers limited enhancement to the textual features, providing only modest complementary information. This limitation is evident in Table 3.
>
>     Superclasses serve as high-level semantic information for establishing relationships between base and novel classes. The purpose is to add shared knowledge between categories to text embeddings belonging to the same superclass, thereby enhancing the generalization ability to novel classes. Unlike using LLM to generate only the rich information contained in the categories. Therefore, the amount of information contained cannot be used to evaluate the contribution of the superclass in hierarchical prompts. Meanwhile, although Table 3 shows the limitations of hierarchical prompts, it still steadily improves the open vocabulary detection performance of the baseline on novel and base classes. Furthermore, HiCA has different performance on different baselines. We supplement the comparison experiment of HiCA with BARON as the baseline, and the results are shown as follows (Tables 1 and 2 in the paper). As shown in the table, the $\mathrm{mAP}_N$ of HiCA is improved by 3.1%, and the $\mathrm{AP}_r$ of HiCA is improved by 1.5%.
>
>    |||||||
>    |-|-|-|-|-|-|
>    |Methods|Detector|Prompts|$\mathrm{mAP}_{50}$|$\mathrm{mAP}_B$|$\mathrm{mAP}_N$|
>    |$\mathrm{BARON}$|FRCNN|L(cat)|48.9|54.6|32.9|
>    |$\mathrm{BARON + HiCA(Ours)}$|FRCNN|H(cat+sup)|**53.6**|**59.8**|**36.0**|
>
>
>    ||||||||
>    |-|-|-|-|-|-|-|
>    |Methods|Detector|Prompts|$\mathrm{AP}$|$\mathrm{AP}_c$|$\mathrm{AP}_f$|$\mathrm{AP}_r$|
>    |$\mathrm{BARON}$|FRCNN|L(cat)|29.1|28.9 |31.9|22.8 |
>    |$\mathrm{BARON + HiCA(Ours)}$|FRCNN|H(cat+sup)|32.3|**34.1**|37.0|**24.3**|
>
>
> 3. For open-vocabulary detection, the critical metric is AP-novel. In the experiments, both modules show only limited improvements in this essential metric.
>
>    Open vocabulary detection methods should indeed pay more attention to the performance in novel classes. In the previous experiments, although the improvement of HiCA on novel classes is limited, it still can steadily improve the detection ability for novel categories. By analyzing it down to the categories, we find the performance degradation on individual categories limits the average performance of HiCA on the overall novel class. Meanwhile, the performance of HiCA is also limited by the selection of baseline. Since the OADP itself is not comparable to SOTA, HiCA has not been able to achieve results close to SOTA in the novel class. To this end, HiCA as a plug-and-play method should further explore the impact of baseline. BARON is used as the new baseline to compare with other OVD methods using prompts on OV-COCO and OV-LVIS (Table 1 and Table 2 in the paper). The best performance of 36.0% $\mathrm{mAP}_N$ and 24.3% $\mathrm{AP}_r$ is obtained on novel classes, respectively. Experiments show that HiCA can perform better when combined with a suitable high-performance baseline.

---

### Meta-Review · Area_Chair_oRe7 · 2024-12-11

**Metareview:**

This paper proposes Hierarchical Prompts with Context-Aware Calibration (HiCA) to improve open-vocabulary object detection (OVD). HiCA introduces two modules: Hierarchical Prompts (HP) that superclass information to enrich textual representations and establish shared knowledge between base and novel classes; and Context-Aware Calibration (CAC): Refining classification scores by linking contextual image features with category distributions. Experiments on OV-COCO and OV-LVIS demonstrate improvements over baseline methods, particularly in base class performance. While the hierarchical prompts improve base class detection significantly, their impact on novel class detection is modest. The approach does not introduce substantial innovation compared to prior works leveraging textual enrichment or contextual features. Several reviewers pointed out that similar ideas, such as using LLM or superclass information, have been explored in prior research. Performance heavily depends on the baseline method, with significant improvements observed only with higher-performing baselines like BARON

During the rebuttal, while the authors addressed some questions raised by reviewers, the core concerns on its originality remain. Given these limitations, the submission is not yet ready for acceptance.

**Additional Comments On Reviewer Discussion:**

- Reviewers Ue2F and c8uQ questioned the originality of hierarchical prompts, given prior works leveraging textual enrichment. The authors argued that their method avoids additional data reliance and provides a unique mechanism for knowledge transfer but failed to distinguish their contributions convincingly.
- Reviewers Cb1c, ZtTR, and EcPQ highlighted the limited improvements for novel classes, which are critical in OVD. The authors acknowledged baseline dependency and provided results with BARON, which showed better performance. However, gains remained incremental compared to SOTA.
- Multiple reviewers noted insufficient experimental verification for CAC. The authors added ablation studies demonstrating the impact of cluster size and DG layer depth.

Despite these efforts, the rebuttals did not significantly strengthen the case for acceptance. The reviewers maintained their concerns leading to an overall recommendation to reject.

---

### Decision · Program_Chairs · 2025-01-22

Reject